# Metamodelling of Naturalised Groundwater Levels at a Regional Level in New Zealand

Jing Yang [1,*] , Channa Rajanayaka [1] , Christopher J. Daughney [2], Doug Booker [1], Rebecca Morris [3] and Mike Thompson [3]

1 National Institute of Water and Atmospheric Research (NIWA), Christchurch 8011, New Zealand; channa.rajanayaka@niwa.co.nz (C.R.); doug.booker@niwa.co.nz (D.B.)
2 National Institute of Water and Atmospheric Research (NIWA), Wellington 6021, New Zealand; chris.daughney@niwa.co.nz
3 Greater Wellington, Wellington 6011, New Zealand; rebecca.morris@gw.govt.nz (R.M.); mike.thompson@gw.govt.nz (M.T.)
* Correspondence: jing.yang@niwa.co.nz

**Abstract:** Groundwater is under pressure from increasing demands for agriculture, industry, domestic uses and support of ecosystems. Understanding the natural state of a groundwater system helps policy makers manage groundwater sustainably. Here we developed a metamodelling approach based on stepwise linear regression that emulates the functionality of physically-based models in the three primary aquifers of the Greater Wellington region of New Zealand. The inputs for the metamodels included local weather data, and nearby river flow data. The metamodels were calibrated and validated against the available simulations of naturalised groundwater level time series from physically-based models for 47 selected wells. For 36 of these wells, the metamodels had Nash-Sutcliffe Efficiency and coefficient of determination over 0.5, showing that they could adequately mimic naturalised groundwater level dynamics as simulated by the physically-based groundwater models. The remaining 11 wells had unsatisfactory performance and were typically located far away from rivers or along the coast. The results also showed that modelled groundwater levels in the aquifer's recharge zone were more sensitive to short-term (less than 2 weeks lag) than long-term river flow (above 4 weeks to 1 year lag), whereas the converse pattern was observed for the aquifer's discharge zone. Although some special considerations are needed, this metamodelling framework can be generally applied to other aquifers to support groundwater resource management at a lower cost than updating physically-based models.

**Keywords:** groundwater level; metamodelling; model performance; regional simulation

## 1. Introduction

Groundwater is a very important natural resource and its total volume represents 96% of all earth's unfrozen fresh water [1]. Groundwater is not only the primary source of drinking water for half of the world's population [2] and the source for over 40% of global consumptive water use in irrigation [3], but it also sustains ecosystems by providing water, nutrients and a relatively stable temperature in streams and lakes [4]. However, with combined effects of increasing groundwater abstraction and climate change, there is a global depletion in groundwater resources [5,6]. Therefore, understanding of the natural state and prediction of groundwater resource availability, e.g., water level, is a crucial step to manage groundwater resources for water allocation purposes and increase the resilience under climate variability and change.

To predict groundwater resource dynamics, traditionally three different approaches are used: conceptual modelling [7,8], statistical modelling [9,10] and physically-based numerical modelling (e.g., MODFLOW [11]; FEFLOW [12]; HydroGeoSphere [13]). Conceptual modelling is usually used to model the general water balance (e.g., input to and

output from the groundwater system) and normally provides a basis for physically-based numerical modelling. Statistical modelling usually applies statistical analysis on existing data (e.g., trend analysis). Numerical modelling can give physically sound predictions if the model structure is robust in representing the physical processes accurately and is calibrated satisfactorily; however, it needs great effort in data collection (e.g., geological data), model calibration and model simulation.

This study focuses on the Greater Wellington region, which lies in the southern part of the North Island of New Zealand (Figure 1). Groundwater is particularly important in the region as a supply of water for irrigation, providing 60% of the region's irrigation allocation. Groundwater also provides 30% of the region's public and community water demands [14]. However, over the past two decades, groundwater levels at some wells have shown a declining trend, suggesting a decreasing availability of groundwater over time [15]. Greater Wellington is responsible for managing water resources in the region. Greater Wellington requires an understanding of naturalised groundwater levels to assist with sustainable groundwater management (e.g., status and trend) and to fulfil national regulatory requirements such as state of the environment reporting and limit setting for resource use. Here, naturalised groundwater levels refer to the conditions that would exist in the absence of any human abstractions. Note that the available measured groundwater level data do not represent natural groundwater levels because the groundwater systems have been significantly altered by humans through water extraction.

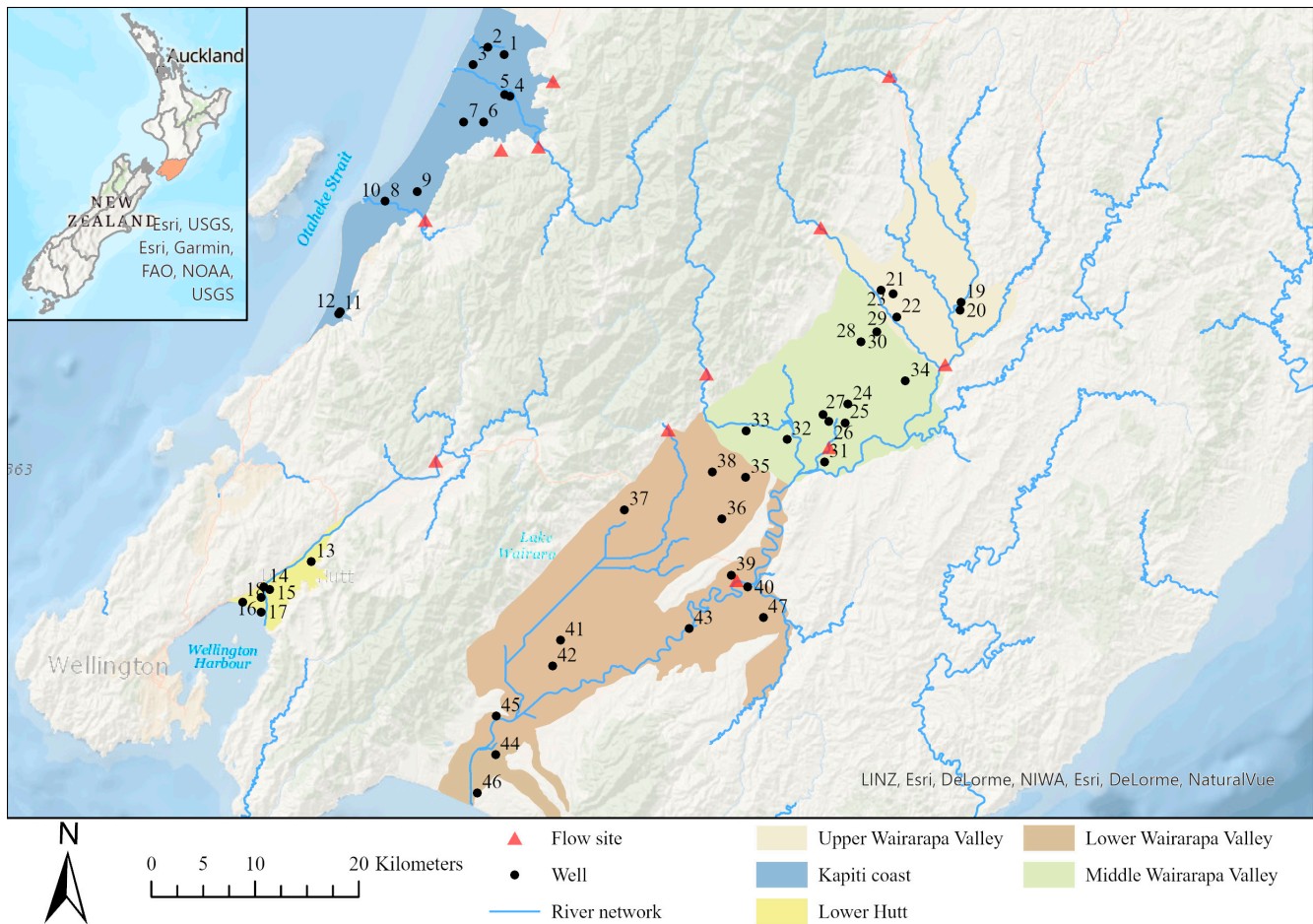

**Figure 1.** Locations of river flow sites (triangles and alphabets) and 47 wells (black dots and numbers) in the Greater Wellington region. Numbers correspond to well number in the first column of Table 1, and only 7 flow sites were selected for metamodelling. Shaded areas are the aquifers for GWRC's existing physically-based models.

**Table 1.** Site information of 47 wells selected for groundwater level modelling.

| No | Site ID | Well Depth (m)[1] | Risk Category[2] | Aquifer | Flow Site |
|---|---|---|---|---|---|
| 1 | S25/5332 | 9.1 | Low | Kapiti coast | |
| 2 | S25/5329 | 25.3 | Low | Kapiti coast | |
| 3 | R25/5228 | 31.7 | Low | Kapiti coast | A: Otaki at Pukehinau |
| 4 | S25/5228 | Unknown | Low | Kapiti coast | A: Otaki at Pukehinau |
| 5 | S25/5258 | 6.0 | Low | Kapiti coast | A: Otaki at Pukehinau |
| 6 | S25/5208 | 192.0 | High | Kapiti coast | |
| 7 | R25/5123 | 13.0 | Low | Kapiti coast | |
| 8 | R26/6594 | 74.0 | Low | Kapiti coast | B: Waikanae at WTP |
| 9 | R26/6626 | 15.8 | High | Kapiti coast | |
| 10 | R26/6916 | 21.0 | Low | Kapiti coast | B: Waikanae at WTP |
| 11 | R26/6503 | 14.8 | Low | Kapiti coast | |
| 12 | R26/6520 | 6.0 | Low | Kapiti coast | |
| 13 | R27/1117 | 14.4 | NA | Lower Hutt | C: Hutt at Birchville |
| 14 | R27/6386 | 115 | NA | Lower Hutt | C: Hutt at Birchville |
| 15 | R27/1115 | 23.47 | NA | Lower Hutt | C: Hutt at Birchville |
| 16 | R27/0120 | 29.6 | Low | Lower Hutt | C: Hutt at Birchville |
| 17 | R27/0122 | 26.2 | Low | Lower Hutt | C: Hutt at Birchville |
| 18 | R27/0320 | 114.6 | Low | Lower Hutt | C: Hutt at Birchville |
| 19 | T26/0243 | 47.5 | High | Upper Wairarapa Valley | D: Ruamahanga Mt Bruce |
| 20 | T26/0501 | 5.1 | Moderate | Upper Wairarapa Valley | D: Ruamahanga Mt Bruce |
| 21 | S26/0033 | 12.0 | Low | Upper Wairarapa Valley | E: Waingawa at Kaituna |
| 22 | T26/0429 | 9.9 | Low | Upper Wairarapa Valley | E: Waingawa at Kaituna |
| 23 | S26/0030 | 38.0 | High | Upper Wairarapa Valley | E: Waingawa at Kaituna |
| 24 | S26/0738 | 5.4 | Moderate | Middle Wairarapa Valley | |
| 25 | S26/0743 | 33.0 | Very high | Middle Wairarapa Valley | |
| 26 | S26/0568 | 45.0 | Low | Middle Wairarapa Valley | |
| 27 | S26/0675 | 31.5 | High | Middle Wairarapa Valley | |
| 28 | S26/0229 | 23.8 | Low | Middle Wairarapa Valley | E:Waingawa at Kaituna |
| 29 | S26/0236 | 41.4 | High | Middle Wairarapa Valley | E: Waingawa at Kaituna |
| 30 | S26/0242 | 7.5 | Low | Middle Wairarapa Valley | E:Waingawa at Kaituna |
| 31 | S27/0248 | 7.9 | Low | Middle Wairarapa Valley | |
| 32 | S26/0545 | 18.0 | High | Middle Wairarapa Valley | F: Waiohine Gorge |
| 33 | S26/0490 | 5.0 | Moderate | Middle Wairarapa Valley | F: Waiohine Gorge |
| 34 | T26/0326 | 10.0 | Low | Middle Wairarapa Valley | |
| 35 | S27/0202 | 4.8 | High | Lower Wairarapa Valley | |
| 36 | S27/0099 | 16.8 | Low | Lower Wairarapa Valley | |
| 37 | S27/0012 | 66.5 | Very high | Lower Wairarapa Valley | |
| 38 | S27/0148 | 8.8 | Low | Lower Wairarapa Valley | |
| 39 | S27/0346 | 9.5 | Low | Lower Wairarapa Valley | G: Ruamahanga Waihenga |
| 40 | S27/0381 | 20.9 | Low | Lower Wairarapa Valley | G: Ruamahanga Waihenga |
| 41 | S27/0428 | 43.6 | Very high | Lower Wairarapa Valley | G: Ruamahanga Waihenga |
| 42 | S27/0434 | 45.2 | Very high | Lower Wairarapa Valley | G: Ruamahanga Waihenga |
| 43 | S27/0542 | 19.0 | Moderate | Lower Wairarapa Valley | G: Ruamahanga Waihenga |
| 44 | S27/0594 | 44.0 | Low | Lower Wairarapa Valley | G: Ruamahanga Waihenga |
| 45 | S27/0576 | 55.5 | Moderate | Lower Wairarapa Valley | G: Ruamahanga Waihenga |
| 46 | R28/0002 | 17.0 | NA | Lower Wairarapa Valley | G: Ruamahanga Waihenga |
| 47 | S27/0571 | 32.0 | Low | Lower Wairarapa Valley | G: Ruamahanga Waihenga |

[1] Well depth means the drilled depth of the well; [2] "NA" in the column means "Not Assessed" in the risk study [15].

In the Greater Wellington region, there are three physically-based models for estimating naturalised groundwater levels, corresponding to the three principal aquifer systems that exist within the region [16–20]. Updating and upgrading these physically-based models with recent data such as climate and water abstraction data would be time-intensive and expensive. Despite several recent advances in model-data assimilation approaches, the process of updating numerical groundwater models with new input datasets (e.g., land surface recharge, river flow) is expensive and generally cost-prohibitive to continuously

use in decision-making. Accordingly, in recent years, Greater Wellington has considered alternative and innovative modelling approaches to support rapid decision-making.

The purpose of this study is to evaluate whether a metamodelling approach can mimic the dynamics of the groundwater system in the Greater Wellington region. In so doing, we assess how metamodelling performance varies in terms of location and depth, and what drives the groundwater system spatially. The metamodels used in this study are based on a conceptual model which generally describes the relationships between the groundwater system and the weather and river systems. Metamodels (i.e., a "model to a model") are developed and trained based on existing physically-based models, and can overcome the key disadvantages of the three traditional approaches listed above [21,22]. The metamodelling approach uses existing numerical groundwater models to develop significantly lower-cost estimates of groundwater level for continuous, and potentially for near real-time, decision-making [23–25]. These metamodels, such as Linear Regression and machine learning technology (e.g., Artificial Neural Network—ANN), have been widely used for water resources management including groundwater modelling [25–30]. Despite computational efficiency of metamodelling, there are limitations and challenges in its application, including high-dimensional problems, uncertainties, validation and overfitting, etc. These have been reviewed and addressed in the literature (e.g., see [31] for a detailed review on metamodelling in water resources and [32] for an general introduction on machine learning for hydrologic sciences). In this study, we developed a metamodelling framework by combining a conceptual groundwater model and the multivariable regression method to simulate "naturalised" groundwater levels across the Greater Wellington region to assist with groundwater resource management. Compared with machine learning methods (e.g., ANN) which are often regarded as "black box" and involving "big data", the linear regression method is easy to apply and maintains the physical interpretation of the relationship between inputs and outputs [26,27]; thus, it is suggested to be applied even before using machine learning methods [27].

## 2. Materials and Methods

### 2.1. Methodology

#### 2.1.1. Conceptual Model

Natural groundwater level at a location is driven by various factors. These factors include the amount and timing of local land surface recharge (*LSR*) and/or recharge from river seepage (if relevant), the groundwater flow dynamics of the regional groundwater system, and soil and aquifer hydraulic properties.

The amount and timing of *LSR* into the aquifer system is driven by weather conditions (e.g., precipitation (*P*), temperature (*T*), and potential evapotranspiration (*PET*)), plant water uptake, soil water holding capacity and topography (land slope). Statistically, if there is no change in land cover or land use, *LSR* can be simulated as a function of weather variables, i.e.,

$$LSR = f(P; T; PET; \ldots) \tag{1}$$

River flow, within a groundwater zone, can interact with the groundwater (e.g., through hydraulic connection) as a losing or gaining stream. Therefore, rivers can also impact groundwater levels along the river course where a hydraulic connection exists.

River flow is also driven by the upstream weather conditions. Thus, the river flow can also be simulated as a function of weather variables:

$$Q = f(P_u; \; T_u; \; PET_u; \; \ldots) \tag{2}$$

where $Q$, $P_u$, $T_u$, and $PET_u$ are river flow and upstream precipitation, temperature, and *PET*. "Upstream" here means the catchment area in which water accumulates and then flows to the given flow recorder site.

Inflow and outflow of groundwater at a specific location are controlled by the regional groundwater system, which is also a function of the *LSR*.

To sum up, the groundwater level ($L$) at a location can be simulated as a function of river flow and local weather conditions:

$$L = f(Q; LSR) = f(Q; P; T; PET; \dots) \tag{3}$$

or in a time series form:

$$L_t = f\left(Q_t, \dots, Q_{t-i}; P_t, \dots, P_{t-j}, ; T_t, \dots, T_{t-k}; PET_t, \dots, PET_{t-l}; \dots\right) \tag{4}$$

where $t$ is time, and $i$, $j$, $k$ and $l$ are the time lags between the change in water level and the change in other variables ($Q, P, T$ and $PET$). Variables with time lags have longer-term effects on groundwater level, reflecting surface storage (river), soil storage and groundwater storage. In this study, a time step length of one week was used. It is worth noting that the impact of upstream weather factors ($P_u$, $T_u$, and $PET_u$) are represented by river flow $Q$.

### 2.1.2. Metamodel

Physically-based models (e.g., MODFLOW [11]) and conceptual process models (e.g., TopModel [33]) are often used in hydrology to solve the above equations. One disadvantage of these models, as described above, is that they normally require additional data or assumptions (e.g., hydraulic conductivity) and model calibration.

In this study, the metamodelling approach was applied using a linear regression model to simulate the relationship between groundwater level, river flow and weather conditions (i.e., Equation (3)), for each groundwater well site:

$$L_t = \sum_{i=1}^{N_0} a_{0,i} X_{i,t} + \dots + \sum_{i=1}^{N_k} a_{k,i} X_{i,t-k} + b \tag{5}$$

where $X_{i,t}$ and $X_{i,t-k}$ are the weather or flow variable at time step $t$ and at time step $t - k$ (i.e., with a lag time of $k$ time steps), respectively, and $a_{k,i}$ and $b$ are corresponding regression coefficients.

Generally, not all variables are important in the regression for a given site. Thus, we employed stepwise regression, using the Akaike Information Criterion (AIC) and the Bayesian Information Criterion (BIC), to determine which independent variables needed to be retained in the final model at each site [34].

### 2.2. Case Study and Data

#### 2.2.1. Groundwater Systems in Greater Wellington Region

There are three principal groundwater areas in Greater Wellington region (Figure 1): Kapiti Coast, Hutt Valley and Wairarapa Valley. Groundwater in Kapiti Coast is mainly used for irrigation. Hutt Valley groundwater is a major drinking water source to the Greater Wellington metropolitan area. Water use in the Wairarapa Valley is mainly for irrigation. In total, groundwater use in the region accounts for approximately one-third of the total annual water allocation. Groundwater is particularly important as a supply of water for irrigation, providing 60% of the region's irrigation allocation. Groundwater also provides 30% of the region's public and community water supply allocation [14].

Previously, for each principal groundwater area, a physically-based groundwater model has been developed for groundwater management purposes. The Wairarapa Valley groundwater model (covering a total area of 1073 km$^2$) consists of three sub-regional numerical groundwater flow models: upper valley [17], middle valley [18], and lower valley [19], implemented with the FEFLOW finite element code [25]. The Hutt Valley groundwater model [16] was developed using MODFLOW code [11], covering an area of 29 km$^2$. The Kapiti Coast groundwater model [20] was developing using MODFLOW code, covering an area of 172 km$^2$.

### 2.2.2. Naturalised Groundwater Levels

The naturalised groundwater levels at each well were simulated by the corresponding physically-based groundwater model described in Section 2.2.1. These previous estimates of naturalised groundwater levels were obtained by running the physically-based groundwater models with all abstractions set to zero. Time periods of available naturalised flow predictions from the physical models are 1992~2007 for the Wairarapa Valley, 1992~2019 for the Hutt Valley, and 1992~2007 for the Kapiti Coast.

The modelled naturalised groundwater levels have been used to assess the potential risk to aquifer systems in the region. Greater Wellington identified wells that are potentially at high risk from groundwater level depletion, based on an analysis of groundwater level trends across the three main aquifer systems. The wells were categorised into four broad categories: Very high risk, High risk, Moderate risk, and Low risk. More details can be found in [15]. For this study Greater Wellington selected 47 wells (as listed in Table 1 and Figure 1) which cover all four risk categories (as in column "Risk Category" in Table 1). These 47 well sites were selected to represent a good spatial and depth coverage of the main aquifers in the three primary groundwater zones (Kapiti Coast, Wairarapa and Hutt Valley). They comprise around one third of the total state of environment monitoring network operated by Greater Wellington and are considered to provide a good representation of the variability in overall groundwater conditions in the region as well as aquifer depletion risk from abstraction (very high, high, medium, and low).

### 2.2.3. Weather Data and Flow Data

Weather data were obtained from NIWA's Virtual Climate Station Network (VCSN [35]). The VCSN data are estimates of daily climate variables (such as rainfall, PET, air and vapour pressure, maximum and minimum air temperature, soil temperature, relative humidity, solar radiation, and wind speed) on a regular (~5 km spatial resolution) grid covering the whole of New Zealand. These estimates are produced every day, based on the spatial interpolation of actual data observations at climate stations located around New Zealand (https://niwa.co.nz/climate/our-services/virtual-climate-stations; accessed on 29 August 2023).

River flow data were supplied by Greater Wellington at each flow monitoring site (triangle in Figure 1) which is either close to or upstream of each well along the groundwater flow path. Most flow sites have long-term records and are positioned to represent natural upstream surface runoff.

### 2.3. Application Procedure

Figure 2 illustrates the metamodelling procedure, including data preparation, processing, and metamodelling and assessment, which are elaborated in the following.

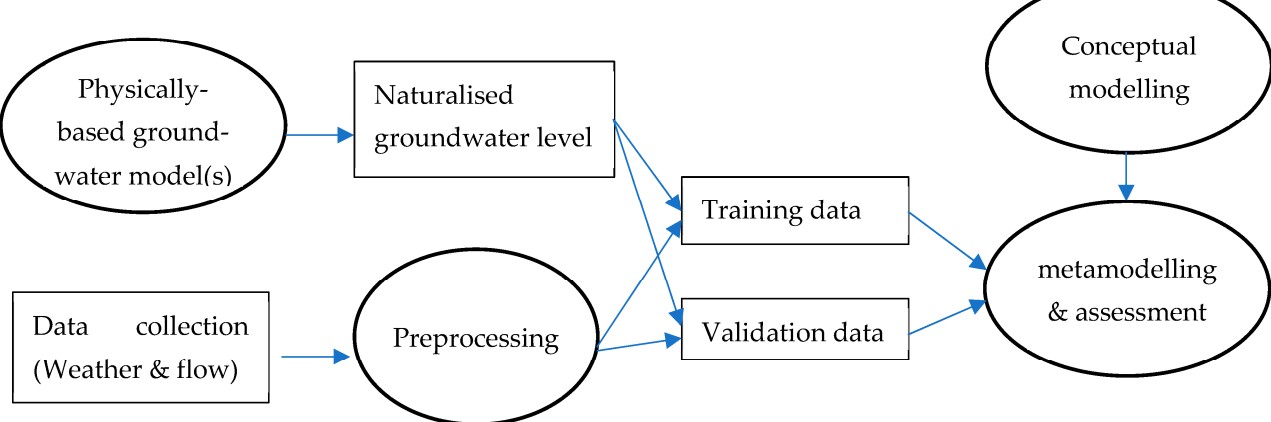

**Figure 2.** Metamodelling application procedure.

### 2.3.1. Initial Assessment of the Groundwater Level Data

The naturalised groundwater level data used for this study are from scenario simulations from the three physically-based groundwater models described above. Notably, the initial setup of the numerical groundwater models had typically influenced the first year of the simulated groundwater levels, which is normally deemed as the "warm-up" period when running numerical models. Therefore, groundwater level data from this warm-up period were not used as input for the metamodelling.

### 2.3.2. Compilation of Weather and Flow Data

Daily P, T, and PET data were extracted for each well site from then VCSN dataset. These daily data were then aggregated to a weekly time step to match the time step of the groundwater level data.

River flow sites and daily flow data assigned to each well site were provided by Greater Wellington. These daily flow data were also aggregated to weekly timesteps.

The weather data were derived from a site close to the given groundwater well, in order to reflect the local weather driver for LSR (Equation (1)). The river flow data were sourced from sites upstream and/or downstream of the given groundwater wells, reflecting the upstream driver and/or downstream hydraulic connection (Equation (2)).

### 2.3.3. Setup of Model Predictors

The following factors were considered as potential model predictors:

- weekly P, T, and PET (referred to as P_1, T_1 and PET_1), lagged weekly P, T, and PET with lag times of 2, 4, 9, 13, 17, 21, 26, 30, 40 and 50 weeks (e.g., lagged P is referred to as P_2, P_4 and so on), shifted weekly P, T, and PET with 2, 4, 9, 13, 17, 21, 26, 30, 40 and 50 weeks (e.g., shifted weekly P is referred to as P_s2, P_s4, and so on). Other weather variables (e.g., humidity) were excluded from a preliminary selection based on correlation analysis to reduce the number of model predictors.
- weekly ratio of P and PET (PoP) with lag times of 2, 4, 9, 13, 17, 21, 26, 30, 40 and 50 weeks (as PoP_2, PoP_4, and so on). PoP is also referred to "dryness" of the weather condition (for a short period) or of climate condition (over a long period).
- weekly flow data (referred to as Flow_1), lagged weekly flow at 2, 4, 9, 13, 17, 21, 26, 30, 40 and 50 weeks (referred to as Flow_2 and so on), and shifted weekly flow at 2, 4, 9, 13, 17, 21, 26, 30, 40 and 50 weeks (referred to as Flow_s2 as above).

Here, 'lagged weekly data at n weeks' means the weekly data averaged over the past n weeks which represent the impact of physical storages (e.g., soil column and river), and 'shifted weekly data at n weeks' means the weekly data n weeks ago which gives the direct impact of the variables as compared to the impact of physical storages. The lag times of increasing intervals (i.e., from 2, to 4, 9, 13, 17, 21, 26, 30, 40 and 50) are to distinguish the short and long term effects (the longer the lag times, the smaller difference between corresponding variables). We chose a maximum lag time of 50 weeks (equivalent to rounded 1 year lag time), reflecting lagged impact of weather and flow on the groundwater within 1 year. However longer than 1 year lag time can also be selected if it reflects the reality, and there is a high correlation in the long term lagged weather and flow variables (e.g., we didn't choose variables with lag time over than 50 which has a correlation higher than 0.8 with the one with lag time 50).

### 2.3.4. Model Calibration, Validation and Assessment

In hydrological modelling, the selection of the calibration period is very important. Ideally, the calibration period should cover a wide range of weather conditions (e.g., wet and dry years). However, one purpose of this study is to extend naturalised groundwater levels to recent years (i.e., beyond the periods covered by the previously developed numerical models). Therefore, we simply used the 75%:25% data split for calibration and validation periods, i.e., the first 75% of the data were used for the calibration period and the rest for

the validation period. The final model was selected through stepwise regression analysis based on Akaike Information Criterion (AIC) and the Bayesian Information Criterion (BIC).

Past literature shows that criteria for model performance assessment are case dependent and there are few generally accepted standards in hydrology, to the best of our knowledge. In accordance with standard procedures in surface water hydrology, the coefficient of determination ($R^2$) and the Nash-Sutcliffe Efficiency (NSE) were used in this study to assess model performance. We didn't use Kling–Gupta efficiency [36], as "unlike NSE, KGE does not have an inherent benchmark against which flows are compared", "Modellers using KGE must be specific about the benchmark against which they compare their model performance" [37], and traditional accumulated knowledge on NSE cannot be directly converted KGE. Criteria listed in Table 2 developed by for river flow were used to assess the goodness-of-fit for the performance of the metamodel, in terms of its ability to match Greater Wellington's previously simulated values for the naturalised groundwater level at each site. These criteria were used in both the calibration and validation periods.

**Table 2.** Performance criteria from Moriasi et al. (2015) [38].

| Performance Metric | Very Good | Good | Satisfactory | Not Satisfactory |
|---|---|---|---|---|
| $R^2$ | $R^2 > 0.85$ | $0.75 < R^2 \leq 0.85$ | $0.60 < R^2 \leq 0.75$ | $R^2 \leq 0.60$ |
| NSE | NSE > 0.80 | $0.70 < \text{NSE} \leq 0.80$ | $0.50 < \text{NSE} \leq 0.70$ | NSE ≤ 0.50 |

## 3. Results

### 3.1. Model Performance

Figure 3 lists metamodel performance indices (NSE) for the calibration and validation periods for the tested sites in all three aquifers. Generally, performance metrics (i.e., $R^2$ not shown) were very similar for the calibration and validation periods, and therefore we opted to classify the performance of the model based on the validation period.

In the Kapiti Coast area, simulations for 7 out of 12 well sites are classified as from "Satisfactory" to "Very Good", indicating the general effectiveness of the metamodelling approach. Most sites with metamodel fits classified as "Satisfactory" to "Very Good" are located close to the main rivers, indicating the influence of both river flow and weather on the groundwater level dynamics. By contrast, there are five well sites with metamodel fits classified as "Not Satisfactory". Two plots in the top left of Figure 4 compare the time series of naturalised groundwater levels (estimated by the physically-based models) versus the metamodels for two sites in Kapiti Coast (S25/5528 and S25/5332). Simulation at S25/5228 (classified as "Good") has good matches to the naturalised groundwater levels, including the dynamics and peaks. At S25/5332 (classified as "Unsatisfactory") the metamodel fails to reproduce the groundwater level in the validation period (especially the magnitude), despite a good match in the calibration period and good matches to the groundwater dynamics in both the calibration and validation periods.

In the Hutt Valley area, all performance metrics (i.e., $R^2$ and NSE) for all wells are classified as "Very Good". This indicates the general effectiveness of the metamodelling approach for matching the simulated naturalised groundwater levels derived from the physically-based model. The bottom-left plot in Figure 4 gives the simulation at R27/1115, corroborating the model performance.

In the Wairarapa Valley area, simulations for 23 out of 29 well sites are classified as from "Satisfactory" to "Very Good", indicating the general effectiveness of the metamodelling approach. Well sites with metamodel fits classified as "Satisfactory" to "Very Good" are located either in the upper valley, in the lower valley, or close to the flow sites. The right three plots in Figure 4 show simulations at three representative sites. These three sites are S26/0033 in the upper valley, S27/0099 in the middle valley, and S27/0434 in the lower valley, which are classified as "Very good", "Unsatisfactory", and "Very good", respectively. Simulations at S26/0033 and S26/0434 have a good match in groundwater level dynamics and peaks (both $R^2$ and NSE are above 0.9 for both calibration period and validation

periods). Although the performance at middle valley ("S27/0099") was classified as "Not satisfactory", the metamodels match the basic dynamics (peak, recession, and valley) of the naturalised groundwater levels, but the magnitudes differ, especially in the peaks. The metamodel estimates are smoother and the peaks are lagged relative to the simulations from the physically-based models. This difference suggests that the naturalised groundwater levels, as simulated by the physically-based models, are affected by one or more drivers or site-specific complexities that are not adequately captured by the formulation of the metamodels. All these "Not satisfactory" sites are located in the middle valley.

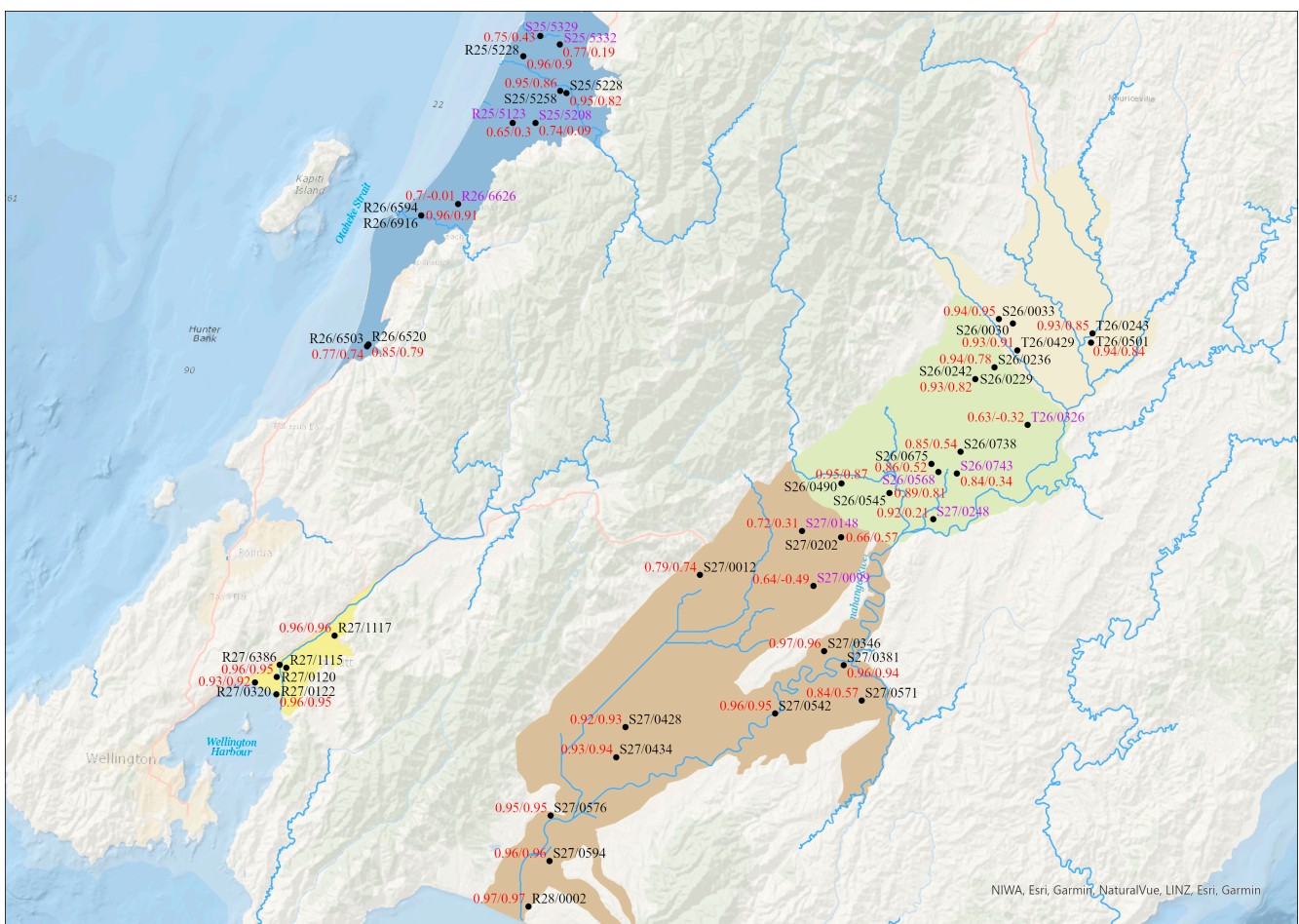

**Figure 3.** Spatial distribution of metamodel performance for the calibration and validation periods. Number "A/B" shows NSE values with A for calibration period and B for validation period. Black site names indicate metamodel fits classified as from "Satisfactory" to "Very Good", while purple site names indicate metamodel fits classified as "Not Satisfactory" for the validation period.

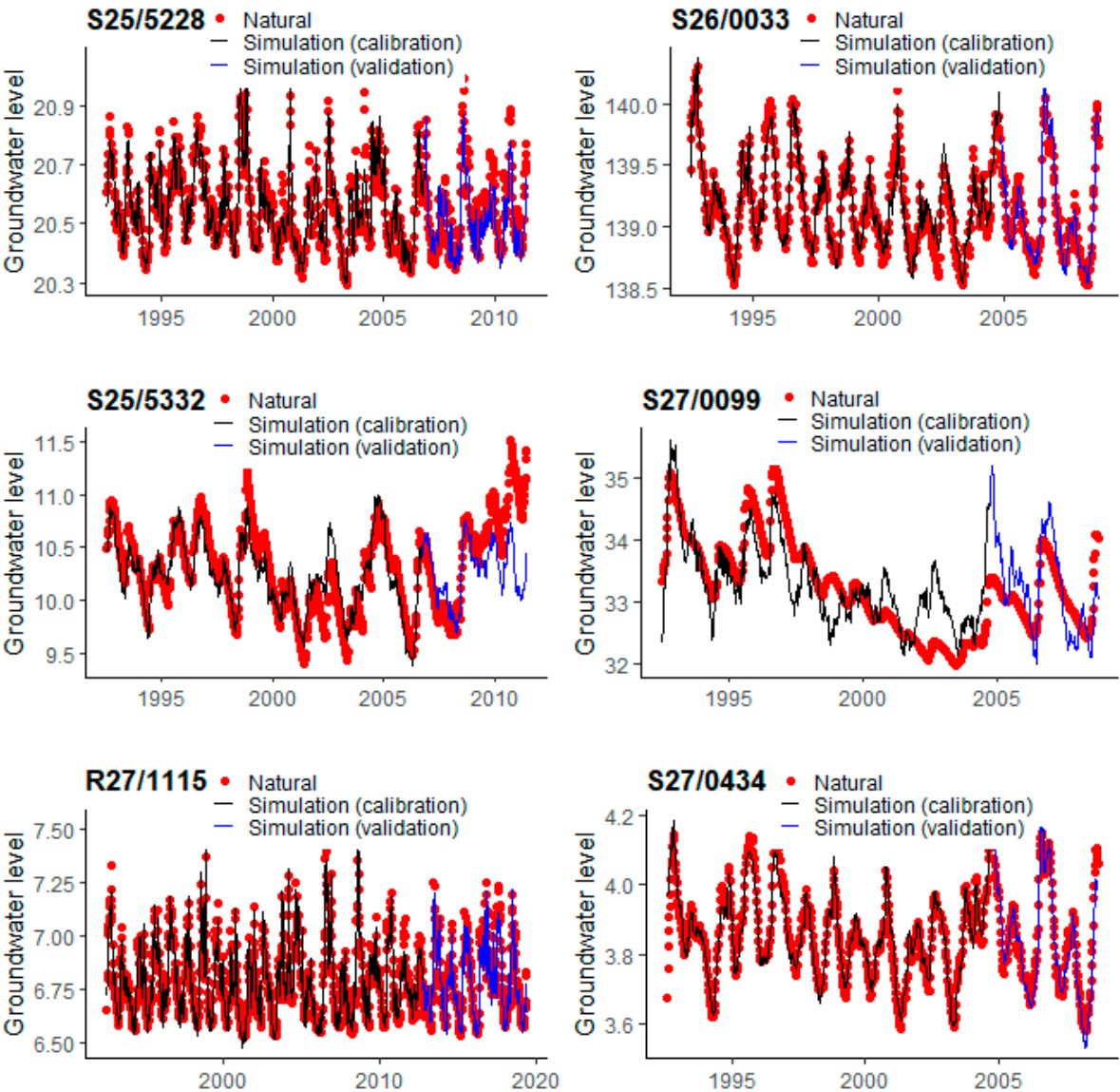

**Figure 4.** Comparison of naturalised groundwater levels estimated from physical-based models versus metamodels at 6 selected sites.

### 3.2. Influential Factors

Figure 5 shows the t statistic of the regression analysis for the 6 selected sites discussed in Section 3.1.

In the Kapiti coast (S25/5228 and S25/5352), the influential factors to groundwater level vary spatially. At S25/5228, the most two influential factors are 'PoP_26' and 'Flow_1', where 'PoP_26' represents the dryness in the past half-year and 'Flow_1' the river flow condition in first week. This indicates that half-year dryness (potentially influencing LSR) significantly impacts the groundwater level, and the interaction with the river is rapid. The impact of precipitation, temperature, and dryness has an over-month lag effect, and the impact of flow is fast (within a week) and can last for up to approximately 1 year (as indicated by Flow_50). At S25/5352, the most influential factor is 'T_50' (i.e., the average temperature in the past year), indicating the importance of yearly average warmness. Compared to S25/5228, the lesser fluctuation in groundwater level time series at S25/5332 (Figure 4) reflects the longer-term impact of weather variables over a period of one month or longer.

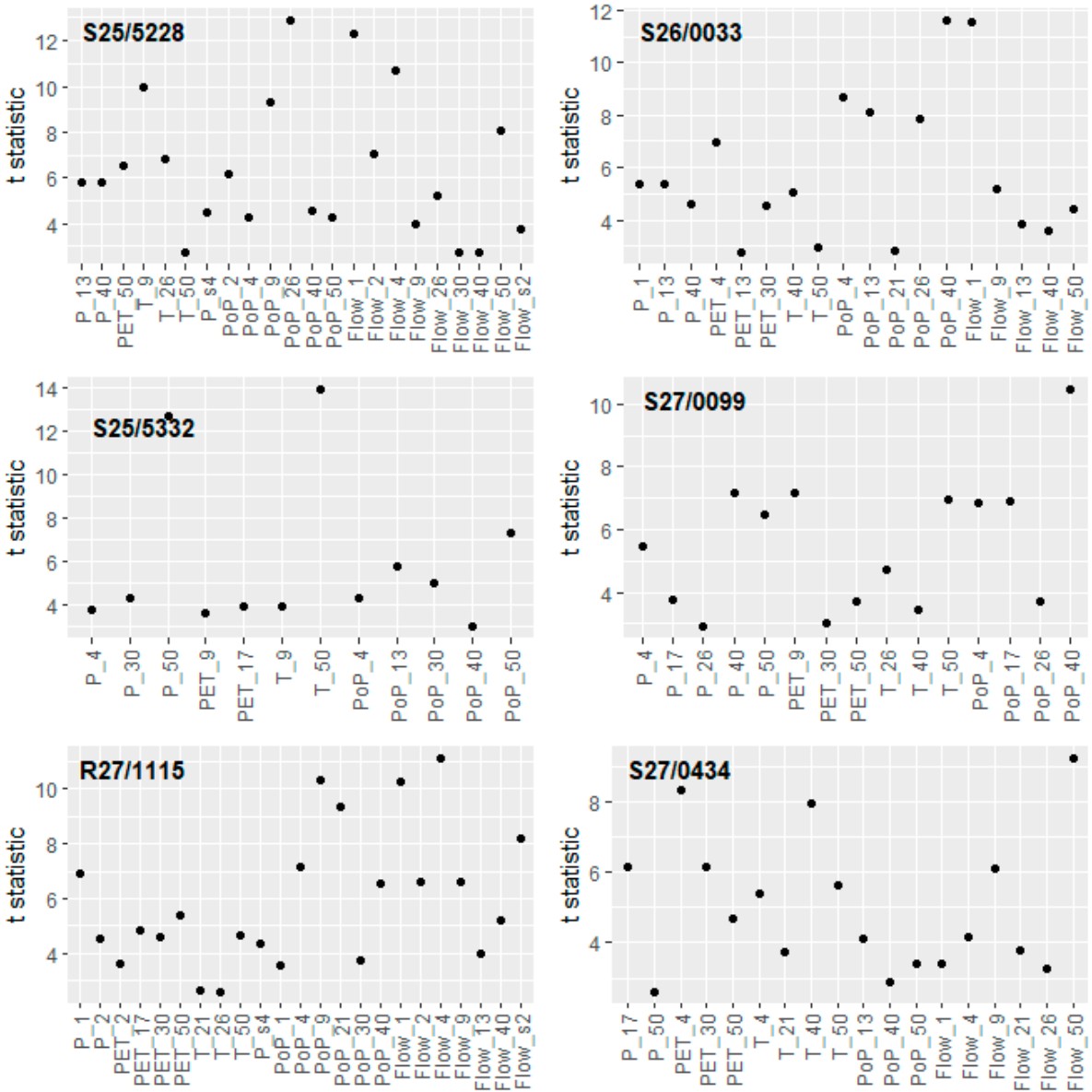

**Figure 5.** t statistic in linear regression analysis of each independent variables at selected sites.

At R27/1115 (the selected site in Hutt Valley), the most influential factors are "Flow_1", "Flow_4", "PoP_9 and "PoP_21", indicating that the interaction with river flow is also fast (in the order of a week) and lasts for about 4 weeks, and the strong response to dryness can be over 2 months. All weather variables play an important role in the fluctuation of groundwater level, among which the response to rainfall is fast and short-lived (within 2 weeks) while the response to other variables can last 1 year.

In the Wairarapa Valley, different factors affect groundwater levels at different sites. At S26/033 (upper valley), the most two influential factors are 'PoP_40' and 'Flow_1'. This indicates that the impact of dryness (a potential function of LSR) has a relatively long lag, i.e., 40 weeks, to groundwater, and the interaction with river flow is very fast (within a week). The impact of precipitation is also rapid and lasts more than half a year, while other climate variables also have an impact. The impact of flow is not only fast (within 1 week) but also lasts up to 1 year, indicating that surface water leakage contributes to slow moving regional groundwater. In the middle valley (S27/0099), the most influential factor is PoP_40, and there is no relation with river flow. Other climate variables have a lag effect over 1 month. In the lower valley (S27/0434), the most influential factor is "Flow_50",

indicating that the yearly flow condition is highly related to groundwater level, although the interaction with flow starts within a week (i.e., Flow_1). The weather variables have a lagged effect on groundwater level over 1 month. The importance of flow on groundwater level is fast (a week) and continues over 1 year, indicating there is a strong interaction with the river.

## 4. Discussion

### 4.1. Model Performance

The metamodel approach provides acceptable fit (from "Satisfactory" to "Very good") at 36 out of 47 well sites covering the three primary aquifers in the Greater Wellington region in New Zealand. The metamodelling performance is not sensitive to well depth (as in Figure 6). Overall, the results demonstrate that, generally, the metamodels can provide good surrogates of naturalised groundwater levels derived from the three physically-based groundwater models previously developed by Greater Wellington.

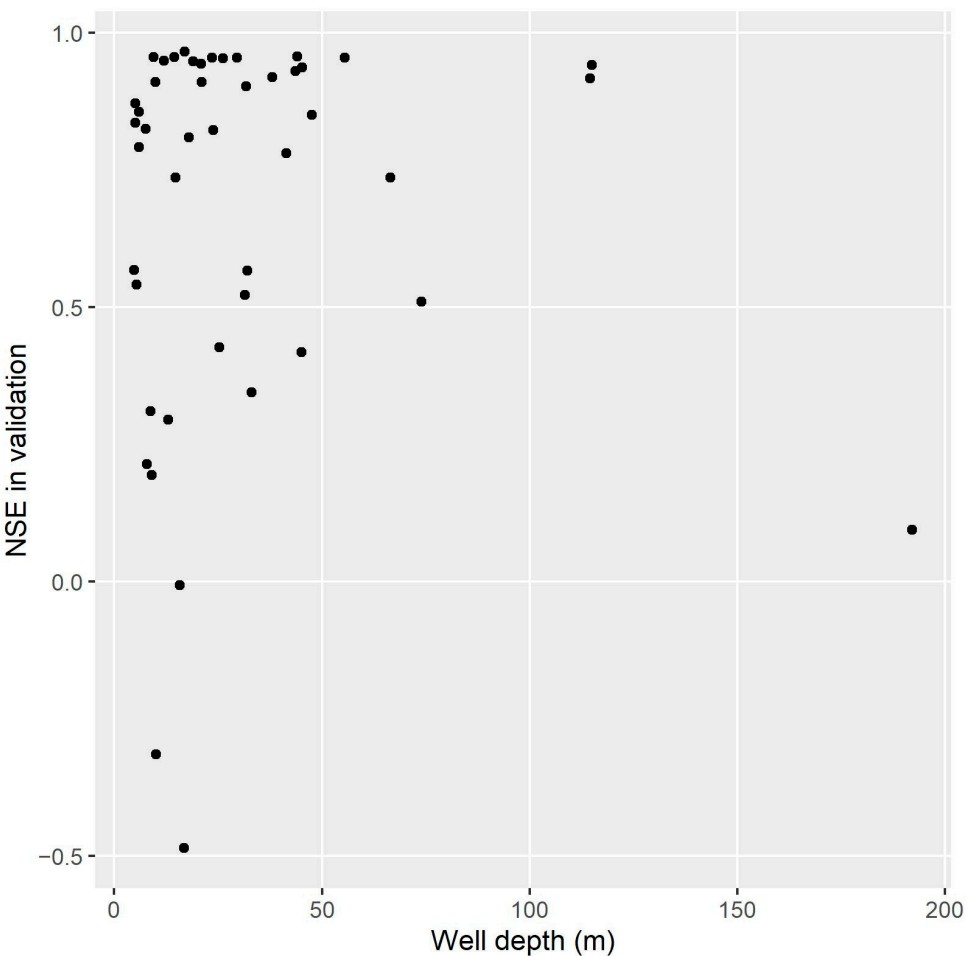

**Figure 6.** Relation of metamodelling performance (in validation) with well depth.

However, there are 11 well sites (out of 47) with metamodel fits classified "Not Satisfactory". These sites are located in the middle Wairarapa Valley and the Kapiti Coast. For the six sites in the middle Wairarapa valley, the metamodel results are mainly sensitive to the weather variables (site S27/0099 in Figure 5), with river flow having little influence on model results, as discussed in Section 3. These weather variables mainly include long term weather variables (e.g., rainfall—"P40"; "PoP40") which contribute to the main variation of the groundwater level. "Not Satisfactory" results might arise from missing or incorrect interpretation of the geologic features (e.g., folds and faults in the area as illustrated in [18]) in the metamodel. In the Kapiti Coast area, the five sites classified as "Not Satisfactory" are

located far away from big rivers (although some sites far away from river are classified as either "Very Good", "Good", or "Satisfactory" (e.g., R26/6503 and R26/6520). For these sites, the metamodels are also largely insensitive to river flow. The lack of sensitivity of the metamodels to river flow may arise because the Kapiti Coast is characterised by a long, narrow aquifer in which groundwater levels may be driven less by river flows and LSR and more by ocean and tidal dynamics. Thus, inclusion of tidal dynamics/sea level as a predictor in the model should be investigated to improve metamodel performance. "Not Satisfactory" results in the Kapiti Coast might also arise from missing or incorrect interpretation of the fault in the area (as illustrated in [20]) or other geologic features in the metamodel, similar to these in the Wairarapa Valley.

For both the Wairarapa Valley and Kapiti Coast, metamodel performance may be improved not just by inclusion of additional relevant input variables (e.g., tidal dynamics), but also by allowing for non-linear relationships in the model, although the non-linear relationship has been partially accounted for with different lag times and compound variables (i.e., PoP).

It is also worth noting that the naturalised groundwater level data (used for calibration of the metamodels) were not observed data but simulations from physically-based groundwater models. Therefore, these naturalised data may have biases or inaccuracies, compared to the true natural groundwater levels.

### 4.2. Importance of Factors

Figure 5 shows the different behaviours of groundwater levels reacting to weather variables and flow at 6 selected sites.

In the Kapiti Coast, sites close to the river (e.g., S25/5228) react rapidly to river flow while there is no reaction for sites away from river (e.g., S25/5332). The corresponding time series of groundwater in Figure 4 also indicates that groundwater level is more dynamic (more fluctuating) at S25/5228 than S25/5332, reflecting the longer-time impact from weather variables.

In the Wairarapa Valley, sites in the middle valley (S27/0099) are similar to S25/5332 in the Kapiti Coast, i.e., they are more influenced by long-term weather variables (instead of river flow and recent (i.e., 2 or 4 week) weather variables), as also observed in the time series plot in Figure 4. In the upper and lower valley, although groundwater levels at both sites (S26/0033 and S27/0434) show similar patterns of sensitivity between the river flow and groundwater fluctuations (Figure 4), the mechanism seems different. In the upper valley, river leakage provides the water sources for the groundwater system, whereas it is the discharge zone to river from groundwater in the lower valley, i.e., Flow_1 is more sensitive than Flow_50 at S26/0033 and the converse is true at S27/0434.

## 5. Conclusions

We developed a metamodelling approach based on stepwise linear regression to emulate the naturalised groundwater levels that had been previously simulated using physically-based models in the Greater Wellington region of New Zealand.

The metamodels can adequately mimic the naturalised groundwater level dynamics at most sites as simulated by the three physically-based groundwater models. This is shown by good simulation results: model performances for 36 out of 47 (77%) wells were classified from "Satisfactory" to "Very Good".

The metalmodelling approach can handle the nonlinear groundwater system (regardless well depth), except for some hydrogeologic features (e.g., fold and fault) and the potential impact of sea level in the middle Wairarapa valley and along the Kapiti Coast. This means that the complexity associated with interactions of the aquifer and weather system needs special attention, and some other variables (e.g., sea level) could usefully be included in the metamodelling in the future.

The metamodelling approach can reflect the distinctive relationship between groundwater level with input variables in recharge and discharge zones. For example, in the

recharge zone of the Wairarapa Valley (upper valley), groundwater level is more sensitive to short-term flow than long-term flow, whereas the converse is true for the discharge zone (lower valley).

Although some special attention is needed for some sites, this metamodelling framework can be generally applied to other aquifers by preparing the regional and site- specific weather, river flow and groundwater level data. Thus, we suggest that the metamodelling approach demonstrated in this study can be used generally and transferrably to support groundwater resources management.

**Author Contributions:** Conceptualization, all authors; methodology, all authors; software, J.Y.; validation, J.Y. and C.R.; formal analysis, all authors; investigation, J.Y.; resources, R.M. and M.T.; data curation, J.Y.; writing—original draft preparation, J.Y. and C.R.; writing—review and editing, C.J.D., D.B., R.M. and M.T.; visualization, J.Y.; supervision, C.J.D.; project administration, R.M.; funding acquisition, R.M. All authors have read and agreed to the published version of the manuscript.

**Funding:** This research was funded by the Greater Wellington, New Zealand, grant number PO268319, and the Strategic Science Investment Fund (SSIF) Programmes of Ministry of Business, Innovation and Employment (MBIE) through NIWA, grant number FWWA2308.

**Institutional Review Board Statement:** Not applicable.

**Informed Consent Statement:** Not applicable.

**Data Availability Statement:** Flow data and groundwater level data can be downloaded from https://github.com/yangjniwa/model_groundwaterlevel/tree/main (accessed on 29 August 2023). For weather data, please contact NIWA for authorisation (https://niwa.co.nz/climate/our-services/virtual-climate-stations; accessed on 29 August 2023).

**Acknowledgments:** This work was funded by the Greater Wellington, New Zealand, and the Strategic Science Investment Fund (SSIF) Programmes of Ministry of Business, Innovation and Employment (MBIE) through NIWA.

**Conflicts of Interest:** The authors declare no conflict of interest.

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
