# Peer review of "Metamodelling of Naturalised Groundwater Levels at a Regional Level in New Zealand"

_sustainability, doi:10.3390/su151813393_

Round 1
Reviewer 1 Report
The manuscript titled "Metamodelling of naturalised groundwater levels at a regional level in New Zealand" focuses on the development of a metamodelling approach using stepwise linear regression to emulate naturalised groundwater levels in the Greater Wellington region of New Zealand. The authors aimed to provide a cost-effective alternative to physically-based models for groundwater management. The abstract provides a clear overview of the study, highlighting the importance of understanding groundwater systems for sustainable management. In the conclusion, the authors summarize their findings and discuss the performance of the metamodels. The authors emphasize the potential application of the metamodeling framework to other aquifers for groundwater resource management. Overall, the manuscript seems to provide a valuable contribution by proposing a metamodelling approach for groundwater level simulation. However, there are some questions:
1. Could you provide more details about the selection criteria for the 47 well sites used in the study? How representative are these sites of the overall groundwater conditions in the Greater Wellington region?
2. In the discussion, you mentioned that the metamodels show good performance for most well sites but have unsatisfactory fits for some locations in the middle Wairarapa Valley and the Kapiti Coast. What are the possible reasons for these discrepancies? Are there specific geological or hydrological characteristics in these areas that might affect the metamodels' performance?
3. The discussion suggests that weather variables play a significant role in the metamodels' performance, particularly in the middle Wairarapa Valley. Could you elaborate on which specific weather variables were considered and how they contribute to the metamodels' accuracy in capturing groundwater levels?
4. Regarding the sites on the Kapiti Coast that exhibited poor metamodel fits, you mentioned that river flow data had little significance. Have you considered other factors that might influence groundwater dynamics in this area, such as tidal dynamics or sea level? If so, how do you plan to incorporate these factors into future iterations of the metamodeling framework?
5. In your conclusion, you mentioned the potential application of the metamodeling framework to other aquifers. What are the key considerations or modifications that would be necessary when applying the approach to different aquifer systems? Are there specific challenges or limitations to be aware of when transferring the methodology to other regions?
6. Considering the cost-effectiveness of metamodeling compared to physically-based models, could you provide some insights into the potential cost savings or resource benefits that could be achieved by implementing the proposed metamodeling approach for groundwater resource management in the Greater Wellington region or other areas?
7. Could you elaborate on the specific weather variables and river flow data that were considered as independent variables in the metamodeling approach? How were these variables selected, and were there any challenges or limitations in obtaining and incorporating them into the analysis?
8. In Equation (5), the linear regression model is used to establish the relationship between groundwater level, weather variables, and river flow. How did you determine the appropriate lag time (k) for incorporating time dependencies in the model? Were there any considerations or insights that guided this decision?
9. Stepwise regression was employed to select the relevant independent variables for each well site based on AIC and BIC. Could you explain the rationale behind using these information criteria and how they helped in determining the final set of independent variables for each metamodel? Were there any cases where certain variables were unexpectedly excluded or included in the final models?
10. Considering that not all variables are important for every site, how did you address potential multicollinearity issues or the presence of redundant variables in the regression models? Were any diagnostic tests or techniques employed to assess the robustness of the selected independent variables?
11. Were there any specific challenges or limitations encountered during the implementation of the metamodeling approach? How did you address issues related to model performance, data availability, or potential biases that might influence the accuracy of the metamodels?
The English language is good.
Author Response
Thanks for the comments. We have addressed these comments in the attachment.

Reviewer 2 Report
The author is congratulated on the extensive numerical work. In this study the authors applied FEFLOW, MODFLOW and Kapiti Coast groundwater models over Greater Wellington region. The authors never discussed about methodology, input data, calibration and validation of these three models. Most part of the paper describes the model validation and calibration process only. No discussion about the model findings or performance. Further without any comparison just presenting the outcome of three numerical models sorry did not make any sense. The objective of paper is quite interesting and however major revision needed before accepting. My observations are given below:
1) Report the literature review and challenges in metamodelling.
2) What are the reason that the authors chosen the linear regression model in 2023? Why can’t incorporated ANN or ML etc.? Possible to generated huge volume of data with the numerical model needed justification.
3) Justify the hypothesis for selecting P, T, PET for predicting LSR and Q. As PET is measured from climatic factors. What is the need of T here?
4) Descriptions about the aquifers system not found. Eg Type, boundary conditions, aquifer parameter, etc.
5) Risk category Colum not found in Table 1.
6) What is well depth in Table 1? Is it water table elevation or draw down or piezometric head. Not able to identified.
7) Table 1 data collected from open well or tube well and period ??
8) What is the temporal resolution of NIWA-VCSN data used in this study?
9) Readers find difficult to follow the methodology section. Plz rework on it. Possibly include flow chart.
10) Just reporting Not Satisfactory, Satisfactory or Very Good are not sufficient to make any decision from the outcome.
11) What are the hypothesis for assigning lag times and shifted weekly intervals? Needs justification.
12) Needs to describe the reason for Not Satisfactory wells(i.e purple site names indicate)?
13) Conclusion just like summary. don’t find any tangible outcome.
Author Response

(The authors gave the same response as above.)

Round 2
Reviewer 1 Report
It can be accepted.
Author Response
Thanks for your feedback.
Reviewer 2 Report
The authors have improved the manuscript in many points compared to the first version. All specific points of criticism have been clarified. However, the few points of criticism have not been eliminated in any way.
1) Most part of the paper describes the model validation and calibration process only. No discussion about the model findings or performance.
2) Further without any comparison just presenting the outcome of three numerical models did not make any sense.
3) Report the literature review and challenges in metamodelling not included.
4) In Table 1 Well depth varies from 4.8 m to 192 m. Hope few wells may be of open wells or tapping an unconfined aquifer. Deeper wells taping confined aquifer. Justify how these extreme changes incorporated in your model development?
5) In Table 1 What are Flow Sit? Describe it.
6) What is good simulations?? In line 391.
7) The lack of sensitivity of the metamodels to river flow may arise because the Kapiti Coast is a characterised by a long, narrow aquifer in which groundwater levels may be driven less by river flows and LSR and more by ocean and tidal dynamics. (Line 393 &394) How the authors clam that ocean and tidal dynamics more sensitive than LSR? How authors clam influenced by tidal dynamics or sea level? Have u done any study on this aspect. Needs physical interpretation.
8) Too many long sentences.. Needs to modify.
9) The results imply that the metamodeling performance is not sensitive to well depth (as in Figure 6), which indicates the metamodel implicitly considers the well depth. (Line 376 &377). What is the hypothesis to select well depth?? Well depth will not play any role if the well not dried up.
10) For the six sites in the middle Wairarapa valley, the metamodel results are controlled primarily by the weather variables (site S27/0099 in Figure 5), with river flow data having little if any significance in the final metamodels, as discussed in section 3. (line 383 & 384). How the authors clam “controlled primarily by the weather variables” “river flow data having little” “if any significance in the final metamodels”. Difficult to understand by readers. Many sentences like this. Needs to modify.
11) T is also selected because PET is a nonlinear function of T and cannot replace T”. PET is also non nonlinear function with humidity, soil temp, sun shine etc. I am not convinced this answer.
12) One or two of the shallower bores are wider diameter (i.e. up to 1-2m) concrete lined wells. None are open well. If diameter of well is more than 1 m it is an open well only. They have larger well storage. How the authors clam is not an open well??
Round 3
Reviewer 2 Report
The authors have improved the manuscript in many points compared to the previous version. However, following comments should be addressed to further improve paper:
1) Still many long and unclear sentences. Needs to correct it. Few examples
Line No 17-19
“Generally, the metamodels can adequately mimic the naturalised groundwater level dynamics as simulated by the physically-based groundwater models, with model performances for 36 out of 47 wells having Nash-Sutcliffe Efficiency and coefficient of determination over 0.5” Not clear that Nash-Sutcliffe Efficiency and coefficient of determination over 0.5
Line No 21-24 (very difficult to understand)
The results also demonstrate that, in a large aquifer with a recharge, transit zone, and discharge zone, groundwater level is more sensitive to short-term (less than 2 weeks lag) than long-term river flow (above 4 weeks to 1 year lag) in recharge zone, while the converse is true in discharge zone
Line No 199-200
“River flow data were supplied by Greater Wellington for a flow monitoring site (triangle in Figure 1) upstream or close to each well” Not clear what is upstream or close to each well. Need to rewrite this sentence.
Line No 290-293 (very difficult to understand)
“Generally, metrics (i.e., R2 not shown) for well sites during the calibration periods are no worse than those in the validation period, and therefore we classified the performance of the model based on the validation period” What is no worse?? How classified the performance of the model?? Rewrite his sentence.
2) In Line 340-341
Explain the hypothecs to identified the most two influential factors. Why limited to two factor? Needed justification
3) In Conclusion Line 440-442
Author has to justify how the fold, fault and sea level affect model performance. They were not considered in model development.
4) Focused conclusion is missing. Requesting to include it.
Still many long and unclear sentences needs to correct it
Author Response
Thanks for your comments and please see our revision and point-by-point response in the attached.
